# Do Reproductive Traits of Invasive Populations of Scotch Broom, *Cytisus scoparius* (Fabaceae), Outperform Native Populations?

**DOI:** 10.3390/plants11162158

**Published:** 2022-08-19

**Authors:** Zigmantas Gudžinskas, Laurynas Taura

**Affiliations:** Institute of Botany, Nature Research Centre, Žaliųjų Ežerų Str. 49, 12000 Vilnius, Lithuania

**Keywords:** habitats, invasive species, ovules, pod size, potential fecundity, realised fecundity, seeds

## Abstract

Reproductive traits are among the most important factors in determining the success of species establishment and invasion in a new area. Studies on transcontinental invasions have revealed that invasive species perform better in the invasive range than in their native ranges. We assumed that the same regularity exists in intracontinental invasions and thus investigated whether the reproductive traits of Scotch broom, *Cytisus scoparius*, perform better in the alien invasive range in Europe compared to its native range in the same continent. The aim of this research was to reveal the potential and realised fecundity of *C. scoparius* in its native and invasive ranges, as well as relationships with the size of pods, habitat type, and how these traits vary at the same site in different years. The results of this study were not able to unambiguously confirm our hypothesis that *C. scoparius* in the invasive range in Lithuania, specifically in the southern part of the Boreal biogeographical region, outperform plants in the native range with regard to the analysed reproductive traits. Potential fecundity of *C. scoparius* in the native range was significantly higher than in the invaded range; however, realised fecundity was not significantly different between the native and invasive ranges. The pod length was similar in both ranges, whereas the pod width was significantly greater in the invasive range than in the native range. The results suggest that the number of ovules per pod, number of matured seeds, and the size of pods are related with the type of habitat and local environmental conditions in the habitat. Although many studies on other species have confirmed higher fecundity and overall better performance of plants in the invasive range than in the native range on other continents, this rule probably cannot be applied for intracontinental invasive species occurring in relatively close geographical regions to their native ranges.

## 1. Introduction

Species traits and their effects as determinants of potential invasiveness have been among the main puzzling questions of invasion ecology. Biological traits affect the invasion success at all life stages of an alien plant, from reproduction to dispersal, and the ability to compete with resident species [1]. Reproductive traits are among the most important characteristics defining the degree of success of a species’ establishment and further spread in a new area [2,3,4]. The potential fecundity in plants usually is estimated as the number of developed ovules, whereas realised fecundity is defined as the number of matured seeds or ratio between the number of ovules and matured seeds [4,5,6]. High fecundity has been recognised as an important correlate of the invasiveness of various plant species [6,7,8,9,10] because it predetermines high propagule pressure, which, in turn, facilitates the spread and further invasion [11,12,13].

The biological traits of plant species and their effect on mechanisms of invasions usually have been compared with native congeners, other phylogenetically related species, or similar non-invasive introduced species [6]. Many studies have been dedicated to analysing various traits of species in their native and invasive ranges, frequently in different continents. Biological and ecological traits, morphological and genetic variation, or ecological performance of species native to Europe, Eurasia, or Asia have been usually studied and compared with their invasive populations in North America [14,15,16,17,18] and Australia [19,20,21]. Similarly, traits and ecological performance of species of North American origin have been compared with their invasive populations in Europe [22,23,24]. Most of the studies confirmed that plant species in the invasive range grow denser, taller, and produce more seeds than in the native range. 

Species originating from other continents usually prevail among the invasive ones with the highest negative ecological impacts [25,26,27]. However, alien plant species originating from different regions of the same continent are also frequent in local floras. For example, as much as 29% of all alien plant species registered in Europe, are native in some regions of the continent [26]. Although alien species originating from the same continent are quite frequent, only a few studies have been dedicated to researching the ecological performance, genetic diversity, and population structure in their native and invasive ranges [28,29,30,31,32]. 

Quite a small fraction of alien species originating from Europe, have been included in national lists of the invasive alien species of European countries, whereas the largest portion of listed species originates from other continents [33,34,35]. Three alien plant species, namely *Cytisus scoparius*, *Gypsophila paniculata*, and *Rumex confertus*, originating from different regions of Europe, are recognized as invasive in Lithuania [36]. The invasive populations of *Cytisus scoparius* in Lithuania are located approximately 300 km north of their native range boundary in Poland. *Cytisus scoparius* currently is the most widespread of these and occupies a wide range of habitats. Although it was introduced in Lithuania at the end of the 19th century [37], its naturalization and fast spread started in the second half of the 20th century, when it was recommended for cultivation in fire protection belts of forests and as a fodder plant for game animals [38,39,40]. 

*Cytisus scoparius* is native to Europe; however, it has become invasive in Asia [41,42], Africa [43], Australia and New Zealand [44,45,46], North America [47,48], and South America [49]. *Cytisus scoparius* has been shown to cause significant declines in native biodiversity around the world [41,42,50,51] mainly by increasing nitrogen content and in some cases decreasing soil pH [52,53,54,55,56]. Although *C. scoparius* typically invades open habitats, it has the potential to establish and survive under conditions of low light availability of the forest understory [44,46]. Thus, it has negative impacts on forestry in natural forests and tree plantations [56], modifies the species composition at invaded sites, and hampers the recruitment of rare and protected plant species, favouring shade-tolerant and alien plant species and the generalist fauna [50,57]. Srinivasan et al. [41] reported that invasion of *C. scoparius* in India negatively impacts grassland community structure and composition, favouring weedy native plants, but does not greatly alter ecosystem functions. However, studies in Lithuania showed that this species in dry pine forests, along forest edges, and in dry grasslands threatens many protected plant species adapted to dry, sunny, and sparsely vegetated grasslands [40]. Its high seed production and long-persistent soil seed bank ensures population recovery and further invasion by *C. scoparius* after natural or artificial elimination of adult individuals [56,58,59,60,61,62]. Although seed production has been exhaustively studied in invasive and to some extent in native populations, the potential and real seed production of *C. scoparius* have only been merely investigated [63]. *Cytisus scoparius* is thought to be an obligate outcrosser and self-pollinated flowers set comparatively fewer seeds than outcrossed flowers [48,64,65]. Since *C. scoparius* relies entirely on seeds for reproduction and is pollinator-dependent, the behaviour of pollinators represents an important aspect of the reproductive biology of this plant species [66]. Therefore, knowledge of the potential and real seed production provides a possibility to forecast seed production under favourable habitat and meteorological conditions and presence of pollinators. 

Considering the significant negative impact of *C. scoparius* in the invaded regions of the world [45,48,49,56,62,67] and its alarming threat to habitats of high conservation value in Lithuania with protected plant species, we initiated studies on its population structure, ecology [39], and reproductive traits. To date, most studies on the reproductive traits confirm higher fecundity of invasive plant species in their invasive range in distant regions of the world compared to their native range [4,8,68]. Therefore, we aimed to test whether this rule can be applied to alien invasive species occurring in regions relatively close to their native range. Following the Baker’s [69] postulate, we hypothesized that the reproductive traits of *C. scoparius* in the invasive range perform better than in its native range. Thus, we formulated the following questions for this research: (a) what is the potential and realised fecundity of *C. scoparius* in its native and invasive ranges? (b) To what extent the potential and realised fecundity depend on habitat type? (c) How does potential and realised fecundity differ in the same habitat in different years? (d) What is the relationship between potential and realised fecundity and the size of pods? 

## 2. Materials and Methods

### 2.1. Study Species

*Cytisus scoparius* (L.) Link (Fabaceae) is a shrub usually growing from 0.5–2.5 m, but occasionally reaching more than 3 m high with several erect or ascending stems and dark green angled branches. It establishes symbiotic relationships with nitrogen-fixing bacteria of the genus *Rhizobium* [52]. The flowers are pedicellate, solitary, or in pairs, born in the axils of leaves. The corolla is golden yellow and 15–25 mm long. *Cytisus scoparius* is thought to be an obligate outcrosser, while self-pollinated flowers sometimes set seed [48,64,65]. Fully developed pods are oblong, strongly compressed, with brown or white hairs, and black at maturity. The seeds are small, ranging from green to brown or reddish brown. Seeds of *C. scoparius* are released from the plant by ballistic explosion of pods [58] into a long-persistent soil seed bank [62,70,71]. 

*Cytisus scoparius* is native to Europe, within an area bounded by Spain and Portugal in the south-west, north to the British Isles, east to southern Sweden and western-central Ukraine, and to the south by northern and central Italy. In other regions of Europe, it is considered as an alien species [72]. *Cytisus scoparius* has become invasive in South Africa, Australia, New Zealand, North and South America, and South Asia [41,44,45,48,49,67]. *Cytisus scoparius* typically occupies open sunny habitats of grasslands and various woodlands [44,46].

### 2.2. Study Sites

Material for the study on the reproductive traits of *C. scoparius* was collected from July to August of 2016. We sampled three populations in the native range of the species in Germany and Luxembourg and nine populations in the invasive range in Lithuania (Table 1, Figure 1). The list of localities was arranged according to the geographical location from south to north. 

The sampling sites were selected to include the main types of woodland, semi-open, and open habitats invaded by *C. scoparius* in Lithuania: grey dunes, thermophile woodland fringes (hereafter, woodland fringes), Scots pine woodlands (hereafter, pine woodlands), taiga woodlands, and shrubby clearings. In the native range, we selected semi-open thermophile woodland fringes and open inland cliff habitats, expecting the highest potential and real fecundity (Table 1). The habitat vegetation characteristics are presented in Appendix A (Table A1). 

Aiming to test the differences in the number of ovules, matured seeds and the pod size in the same site between years, we repeatedly sampled three populations (Juodkrantė, Nagliai and Valkininkai) in Lithuania in 2017, which were studied beforehand in 2016. The mean air temperature in May 2016 and 2017, when *C. scoparius* was flowering, was similar at the study sites (in 2016 at Juodkrantė and Nagliai it was 14.5 °C and at Valkininkai it was 14.2 °C; in 2017 at Juodkrantė and Nagliai it was 12.3 °C and at Valkininkai it was 12.2 °C). Precipitation in May varied more between years and study sites (in 2016, precipitation at Juodkrante and Nagliai was 37.7 mm, at Valkininkai 28.5 mm; in 2017, precipitation at Juodkrante and Nagliai it was 6.3 mm, at Valkininkai 24.0 mm). Data on the temperature and precipitation in the study sites in 2016 and 2017 were taken from the Lithuanian Meteorological Yearbook [73,74]. 

### 2.3. Procedures of Pod Sampling and Study

Stands of *C. scoparius* occurring in a uniform habitat, occupying substantial areas (0.5 ha or more), and containing at least 100 generative individuals were selected for this study. We collected a total of 100 normally developed and ripped intact pods from 50 individuals separated by at least 5 m distance in each site. One pod was sampled from the upper part of the shrub, another from its middle part. Collected pods were placed inside labelled paper bags and brought to the laboratory for further analysis. Length and width of pods were measured using an electronic caliper (*WZSL 150*) with a precision of 0.1 mm. The length of the pod was measured from its tip to the base of the pedicel, the width was measured approximately at the middle of the pod. Each measured pod was opened, and the number of the matured seeds and aborted ovules was counted (Figure 2). 

### 2.4. Statistical Analyses

The normality of data distribution was evaluated using Shapiro–Wilk’s test. The homogeneity of variation (homoskedasticity) of the data was determined by applying Levene’s test. Data sets of the number of ovules, matured seeds, and pod length and width were distributed normally, therefore parametric tests were applied. Presenting descriptive statistics, numerical results are reported as means and standard deviation (± SD). Comparisons of studied traits between populations were performed applying ANOVA tests. Pairwise comparisons between sites were conducted using Tukey’s HSD post hoc test. As data sets from the native range (n = 300) and invasive range (n = 900) were unequal, non-parametric Mann–Whitney U-test was used for their comparison. The realised fecundity in a site was calculated as the ratio between the total number of matured seeds and the total number of ovules in analysed pods. Comparisons between the number of ovules, number of seeds, length, and width of pods in different years at the same sites were performed applying Student’s t-test. To determine the influence of habitat type on reproductive traits, sites were grouped by habitat type, and pooled data were compared. The effect of different survey years on the reproductive traits of *C. scoparius* was assessed by two-way ANOVA. Site was considered as a fixed factor and year as a random factor. Relationships between traits (the number of ovules, number of matured seeds, and pod length and width) were evaluated applying generalized linear model and bivariate linear regression analysis. Our results and data on number of ovules per pod and pod length obtained from references were compared using t-test from the applied parameters. A value of *p* < 0.05 was taken as the level of significance. All calculations were performed employing PAST 4.10 software [75]. 

## 3. Results

### 3.1. Effect of Population Origin and Site on Reproductive Traits

#### 3.1.1. Number of Ovules

The number of ovules per pod from individual sites varied considerably and ranged from 13.65 ± 2.25 to 17.41 ± 1.92 (Table 2). ANOVA test showed significant differences between sites according to the mean number of ovules (F (11, 1188) = 42.54, *p* < 0.001). The largest mean number of ovules was recorded in the native range (the Übereisenbach site), whereas the lowest mean number was recorded in the invasive range (the Meteliai site). A maximum of 24 ovules in a single pod was recorded in the invasive range (the Veisiejai site) and 22 ovules in the native range (the Übereisenbach site and the Vianden site). Pairwise comparisons showed that there are significant differences between certain sites, but the differences cannot be explained by population origin alone (Appendix B, Table A2). 

We found that the mean number of ovules per pod in the native range (17.03 ± 1.81; n = 300) was significantly higher (Mann–Whitney U = 78972.0, *p* < 0.001) than in the invasive range (15.31 ± 2.43; n = 900). These results reject our initial hypothesis that the potential fecundity of *C. scoparius* is higher in the invasive range than in the native range. 

#### 3.1.2. Number of Matured Seeds

The mean number of matured seeds per pod across study sites ranged from 4.72 ± 3.20 to 9.44 ± 3.23 (Table 2). ANOVA test showed significant differences between sites according to the mean number of aborted ovules (F (11, 1188) = 21.78, *p* < 0.001). The smallest number of matured seeds per pod was registered at the Valkininkai site, whereas the largest number was recorded for the Juodkrantė site (Table 2). Both sites are within the invasive range of the species. The maximum of 18 matured seeds in an individual pod was recorded both in the invasive (the Übereisenbach site) and in the native ranges (the Juodkrantė site). Pairwise comparisons of sites according to the number of matured seeds showed a complex relationship that cannot be explained by the origin of the populations, but is suggested to be dependent on habitat type (Appendix B, Table A3). The highest realised fecundity was found at the Juodkrante site (55.24%), while the lowest was at the Valkininkai site (31.28%).

We found that the mean number of matured seeds per pod in the native range (6.43 ± 3.34; n = 300) was lower than in the invasive range (6.58 ± 3.24; n = 900), but the difference was insignificant (Mann–Whitney U = 1.3, *p* = 0.369). These results also reject our initial hypothesis that the realised fecundity of *C. scoparius* is higher in the invasive range than in the native range. 

#### 3.1.3. Number of Aborted Ovules

The mean number of aborted ovules per pod across study sites ranged from 7.31 ± 2.29 to 11.21 ± 2.77 (Table 2). The ANOVA test showed significant differences between sites according to the mean number of matured seeds (F (11, 1188) = 25.38, *p* < 0.001). The smallest number of aborted ovules per pod was registered at the Meteliai site in the invasive range, whereas the largest number was recorded for the Walsdorf site in the native range (Table 2). Pairwise comparisons of sites according to the number of aborted ovules, as in the case of the number of ovules and matured seeds, showed a complex relationship (Appendix B, Table A4).

We found that the mean number of aborted ovules per pod in the native range (10.60 ± 3.01; n = 300) was significantly higher (Mann–Whitney U = 89208,0, *p* < 0.001) than in the invasive range (8.73 ± 2.86; n = 900). Aborted ovules accounted for 62.24% of the total number of ovules in the native range and 57.84% in the invasive range. Thus, the potential fecundity is better realised in the invasive range than in the native range. These results support our initial hypothesis that the realised fecundity of *C. scoparius* is higher in the invasive range than in the native range. 

#### 3.1.4. Pod Size

The mean pod length in the study sites ranged from 41.32 ± 7.43 mm to 49.15 ± 7.49 mm (Table 2). The ANOVA test of the mean pod length showed significant differences between sites (F (11, 1188) = 9.44, *p* < 0.001). The smallest mean pod length was recorded from the Kaltanėnai site, whereas the largest mean pod length was from the Grūtas site (Table 2). Pairwise comparisons did not reveal any clear trends between sites regarding pod length (Appendix B, Table A5). Analysis of the data showed that the mean pod length in the invasive range (44.48 ± 7.41; n = 900) was slightly larger than in the native range (43.77 ± 6.22; n = 300), but the difference was insignificant (Mann–Whitney U = 1.3, *p* = 0.262). 

The mean pod width obtained from the study sites ranged from 8.84 ± 1.35 mm to 10.44 ± 1.10 mm (Table 2). The ANOVA test showed significant differences between the sites regarding mean pod width (F (11, 1188) = 17.88, *p* < 0.001). The smallest mean pod width was recorded from the Juodkrantė site, whereas the largest mean pod width was from the Grūtas site (Table 2). Pairwise comparisons showed that the Grūtas and Juodkrantė sites are significantly different from most other sites by mean pod width (Appendix B, Table A6). The mean pod width in the native range (9.54 ± 0.99 mm) was significantly smaller (Mann–Whitney U = 120510.0, *p* = 0.005) than in the invasive range (9.76 ± 1.20 mm). 

A weak but significant correlation was found between the number of ovules and matured seeds in a pod (r = 0.47, *p* < 0.001). According to the results of the bivariate linear regression analysis, the number of ovules in a pod relates to 22.3% of matured seeds in a pod (r^2^ = 0.223, *p* < 0.001). A moderate relationship was found between the pod length and the number of ovules (r = 0.61, *p* < 0.001) and the pod length correlates to 36.8% with the variation in the number of ovules (r^2^ = 0.368, *p* < 0.001). The same relationship was found between the pod length and the number of matured seeds (r = 0.52, *p* < 0.001; Figure 3) and pod length correlates to 26.7% with variation in matured seed number (r^2^ = 0.267, *p* < 0.001). The relationship between pod length and width was weak but reliable (r = 0.40, *p* < 0.001) and pod length correlates to 16.3% with variation in width (r^2^ = 0.163, *p* < 0.001; Figure 3). However, very weak relationships were revealed both between pod width and the number of ovules (r = 0.12, *p* < 0.001) and pod width with the number of matured seeds (r = 0.18, *p* < 0.001).

### 3.2. Effect of Habitat Type on Reproductive Traits

#### 3.2.1. Number of Ovules

The largest mean number of ovules per pod was revealed in grey dune habitats (16.88 ± 1.88) in the invasive range and in inland cliff habitats (16.84 ± 1.73) in the native range (Figure 4; Appendix C, Table A7). The smallest mean number of ovules per pod (14.39 ± 2.29) was found in western taiga habitats in the invasive range. The ANOVA test of the mean number of ovules showed significant differences between habitat types (F (5, 1194) = 45.70, *p* < 0.001). Significant differences of the number of ovules per pod were found between most pairs of the analysed habitats (Figure 4). The mean number of ovules per pod in shrubby clearing, pine woodland, and western taiga was significantly lower than in inland cliff, grey dune, and woodland fringe habitats (Figure 4). Thus, the complex of ecological conditions of the habitat influences the potential fecundity of *C. scoparius*.

#### 3.2.2. Number of Matured Seeds

The largest mean number of matured seeds was recorded from shrubby clearing (7.73 ± 3.09) habitats, whereas the lowest number of matured seeds per pod was found in pine woodland (5.04 ± 2.93) habitats (Figure 4). The ANOVA test of the mean number of matured seeds revealed significant differences between habitat types [F (5, 1194) = 31.51, *p* < 0.001]. Pairwise comparisons of habitats by the number of matured seeds per pod showed that there were no significant differences between the grey dune and shrubby clearing habitats, nor between the woodland fringe and shrubby clearing habitats (Figure 4). Unexpectedly, in the open inland cliff habitat, the number of mature seeds per pod was not significantly different from that of *C. scoparius* growing in the forest, pine woodland, and western taiga habitats. These results suggest that the cover of trees in the habitat is not a determinant of the realised seed production.

#### 3.2.3. Number of Aborted Ovules

The largest mean number of aborted ovules per pod was recorded from inland cliff (11.11 ± 3.07) habitats, whereas the lowest number of aborted ovules per pod was found in shrubby clearing (7.54 ± 2.36) habitats (Figure 4). The ANOVA test of the mean number of aborted ovules revealed significant differences between habitat types (F (5, 1194) = 37.14, *p* < 0.001). Pairwise comparisons of habitats by the number of aborted ovules per pod showed that there were significant differences between most of habitat types, except for the grey dune and western taiga, grey dune and woodland fringe, western taiga and woodland fringe, and pine woodland and woodland fringe habitats (Figure 4). The highest proportion of aborted ovules was found in the pine woodland (66.03%) and inland cliff (65.97%) habitats, while the proportion was much lower in the shrubby clearing (49.38%) and grey dune (51.95%) habitats. In the woodland fringe (57.64%) and western taiga (59.83%) habitats, the proportion of aborted ovules was intermediate between the other habitats. This suggests that realised fecundity depends on habitat type, but does not explain which factors or complexes of factors have the most important impact.

#### 3.2.4. Pod Size

The largest mean pod length was found in shrubby clearing (45.89 ± 7.51 mm) and in grey dune (45.40 ± 6.80 mm) habitats. The shortest pods were in western taiga (43.05 ± 7.29 mm) habitats (Figure 4; Appendix C, Table A7). The ANOVA test of the pod length revealed significant differences between habitat types [F (5, 1194) = 37.14, *p* < 0.001]. Pairwise comparisons of *C. scoparius* pod length showed that there were no significant differences between woodland fringe and all other habitat types. In the open inland cliff habitats of the native range of the species, the mean pod length was significantly lower than in the open grey dune and shrubby clearing habitats, but was not significantly different from other analysed habitats (Table 3).

The largest mean pod width was found in shrubby clearing habitats (10.36 ± 1.07 mm), whereas the lowest mean pod width was recorded in grey dune (9.06 ± 1.12 mm) habitats (Figure 4). The ANOVA test of the pod length revealed significant differences between habitat types (F (5, 1194) = 32.48, *p* < 0.001). Pairwise comparisons of habitats by pod width showed that in shrubby clearings pods were significantly wider, whereas in grey dune habitats they were significantly narrower than in all other studied habitat types (Figure 4).

### 3.3. Comparison of Reproductive Traits between Years

Analysis of data on reproductive traits of *C. scoparius* collected in 2016 and 2017 revealed some patterns of variation. Mean number of ovules per pod (n = 300) pooled from all sites was 16.28 ± 2.26 in 2016, whereas in 2017 it was 16.70 ± 1.88 and significant differences (t = 2.44, *p* = 0.015) between the study years were found (Table 3). At the level of individual sites, the number of ovules in Valkininkai was significantly larger in 2017, whereas in Nagliai it was larger in 2016. The difference of the number of ovules per pod was insignificant between two years at the Juodkrantė site (Table 3). 

The mean number of matured seeds per pod (n = 300) from three pooled sites in 2016 was 6.96 ± 3.46, whereas in 2017 it was 9.56 ± 3.10 and the difference was significant between the two study years (t = 13.00, *p* < 0.001). By comparing the number of matured seeds in the study sites we found the same regularity as it was with the number of ovules per pod. Significant differences of the number of matured seeds per pod between years were found at the Valkininkai and Nagliai sites, but differences within the Juodkrantė site were insignificant between years (Table 3). Furthermore, the mean number of matured seeds at the Valkininkai and Nagliai sites in 2017 was much larger than it was in 2016. It should be noted that the percentage of ovules having matured to seeds also differed significantly between the study years. 

In 2016, the realised fecundity in all three studied sites pooled was 42,7%, whereas in 2017 it was 57.0%. The largest difference was at the Nagliai site, where the realised fecundity was 40.2% in 2016 and 57.7% in 2017. The largest realised fecundity among all studied sites and years was found in the Juodkrantė site, where the value was 59.1% in 2017.

Analysis of the pod size variation between the two study years showed that mean pod length (n = 300) from all pooled sites in 2016 was 44.94 ± 7.54 mm, and in 2017 it was 46.46 ± 6.00 mm (Table 3). The mean pod length significantly differed between the two years (t = 2.75, *p* = 0.006). The mean pod length in 2017 was significantly larger than in 2016 at Nagliai, but no significant differences of pod length were found at the Valkininkai and Juodkrantė sites between the two years (Table 3). The mean pod width (n = 300) in all pooled populations in 2016 was 9.29 ± 1.78 mm and in 2017 it was 9.62 ± 1.06 mm. Thus, pods in 2017 were significantly wider (t = 3.59, *p* < 0.001) than in 2016. For the individual study sites, pods in Juodkrantė and Nagliai in 2017 were statistically significantly wider than in 2016 (Table 3), but no differences were found between the two years at Valkininkai site.

The results of the two-way ANOVA revealed that site (fixed factor) did not have a significant effect on any of the *C. scoparius* reproductive traits analysed (Table 4). Year (random factor) had a significant effect on the number of matured seeds and the number of aborted ovules per pod. The interaction between location and year had a significant effect on all the reproductive traits of *C. scoparius* examined (Table 4).

## 4. Discussion

### 4.1. Effect of Population Origin and Site on Reproductive Traits

Designing this study, we hypothesized that *C. scoparius* performs better in the invasive range than in the native range with regard to its reproductive traits as it has been postulated by Baker [69] and established by other studies related to invasive species traits in the native and invaded ranges [15,18,21]. However, the results of this study revealed no clearly increased reproductive performance in the invasive range in the southern part of the Boreal biogeographical region of Europe but showed certain relationships of the reproductive traits with habitat types and meteorological conditions. 

Although *C. scoparius* has been included in many studies in the native and invasive ranges [20,45,46,70], data on the number of ovules per pod and variation of this reproductive trait are very scarce. Thus, a direct comparison of our results with those of other studies is not possible. The number of ovules has been studied in Spain and the established mean number of ovules per pod was 13.91 ± 2.08 [63]. Applying the t-test from parameters showed that the number of ovules per pod in two sites from Lithuania (Meteliai and Kaltanėnai) and Spanish localities revealed no significant differences, whereas all other sites from Lithuania, Germany and Luxembourg had significantly higher numbers of ovules per pod. According to the results of this study, the potential fecundity of *C. scoparius* in the native range was significantly higher than in the invasive range in the southern part of the Boreal biogeographical region of Europe. 

Seed production of *C. scoparius* plays an essential role for its further spread and formation of the soil seed bank to ensure long-term population survival [70,71]. Although many studies estimated seed production of individual *C. scoparius* shrubs (e.g., [60,70]), the number of matured seeds per pod is understudied. López et al. [63] have reported that the mean number of matured seeds per pod in Spain was 6.01 ± 4.24. In three of our studied sites, in Juodkrantė, Grūtas, and Übereisenbach, the mean number of matured seeds was significantly higher (Table 2) than the reported values from Spain. No significant differences were found between the means of matured seeds in the four sites, Meteliai, Kaltanėnai, Veisiejai, and Walsdorf, and the reported values from Spain, whereas in the other four studied sites, the number of matured seeds per pod was significantly lower than in Spain. Although the mean number of matured seeds per pod was significantly higher in the native range than in the invasive range, the number of matured seeds at individual sites of the invasive range was significantly higher than in the native range. These results suggest that realised fecundity depends not on the geographical location of a population, but on habitat conditions and the effectiveness of flower pollination.

This study revealed that pod length in the native range and in the invasive range in the southern part of the Boreal biogeographical region of Europe was similar. Information about the pod sizes in other regions is very scarce. The mean pod length of *C. scoparius* in Spain was reported as 37.36 ± 8.84 mm [63]. Thus, pods in Spanish populations were significantly shorter than those investigated from the native range in Germany and Luxembourg as well as from the invasive range in Lithuania (Table 2). Furthermore, the shortest pods recorded during this study were at the Kaltanėnai site, but their mean length was statistically significantly (t = 9.84, *p* < 0.001) greater than the reported value for Spain. However, we found no published information about the pod width of *C. scoparius* from other regions of its native and invasive ranges, and thus comparisons of our results with the those of other studies are impossible. 

### 4.2. Effect of Habitat Type on Reproductive Traits

Analysis of the effect of habitat types on the reproductive traits of *C. scoparius* revealed several important regularities. The highest number of ovules per pod was recorded for plants occurring in open habitats irrespective of their geographic location and origin. The number of ovules per pod in grey dune habitats on the Baltic coast and in inland cliff habitats in Central Europe was significantly higher than in pine forest, woodland fringe, shrubby clearing habitats and taiga woodlands, both in the native and invaded ranges of the species in Europe. The number of ovules per ovary and general fecundity of a plant is predetermined by multiple environmental factors, effects of the previous growth season, and flower bud formation, as well as individual characteristics of the plant [76,77]. The results of our study suggest that light availability in the habitat is the most important factor affecting the number of ovules per pod. In shaded habitats, such as taiga woodland, pine woodland, and woodland fringe habitats, the potential fecundity was significantly lower than in open grey dune and inland cliff habitats. The presumption that the light availability has a strong effect on the number of ovules per pod was indirectly supported by the fact that in different years in the same habitat the mean number of ovules varied only slightly. The importance of light availability on the potential fecundity has been confirmed by studies on several other plants (e.g., [77,78]).

Analysing the number of matured seeds per pod between habitat types revealed a different pattern of variation than that found for the number of ovules per pod. The highest number of matured seeds was registered in shrubby clearing habitats in the invasive range, and it was higher than in grey dune and inland cliff habitats both, in the native and invasive ranges (Figure 4; Appendix C, Table A7). Thus, conditions in shrubby clearing habitats were more favourable for seed development and maturation than in other habitats. The smallest number of matured seeds was recorded for taiga woodland and woodland fringe habitats, although significant differences between these types of habitats were also evident. 

Although the number of ovules per pod in different habitat types was different, this fact also does not explain differences in the number of matured seeds. We assume that the occurring differences are also a result of varying flower pollination success [66,79]. Bumblebees visit flowers of *C. scoparius* more intensively in open habitats (e.g., shrubby clearings, grey dunes) than in forest habitats (e.g., taiga woodlands, pine woodlands), as forests are an obstacle for many bumblebee species occurring in the Boreal region of Europe [80]. In our opinion, different pollination success could explain the significant differences between the number of matured seeds and overall realised fecundity recorded in the two close sites of Juodkrantė and Nagliai. In both these sites, *C. scoparius* occupies grey dune habitats. However, the Juodkrantė site is in an open dune area close to the seashore, whereas the Nagliai site is located in a grey dune habitat surrounded from all sides by Scots pine forest stands. Thus, the higher number of matured seeds and, therefore, higher realised fecundity was in an open habitat because of the more effective pollination of flowers. It is also known that the ratio between the potential and realised fecundity depends not only on the efficiency of flower pollination [66,79], but also on competition among developing seeds in a pod for limited resources [81,82], production of phytohormones, and other internal factors leading to selective seed abortion [83,84]. However, the total number of developed pods on individual shrubs were not evaluated in this study, but this factor might affect the number of matured seeds. 

The analysis of pod length and width variation depending on habitat type revealed certain regularities and these compared positively to the number of ovules and number of matured seeds per pod (Figure 4; Appendix C, Table A7). The largest mean pod length was in open grey dune and shrubby clearing habitats, whereas shortest were in shaded habitats of taiga woodland habitats. The mean length of pods of *C. scoparius* occurring in inland cliff habitats of the native range was similar to the pod length in semi-shaded woodland fringe and shaded pine woodland habitats. These results suggest a significant effect of light availability and intensity in habitats on the reproductive traits [77,78]. However, the pod width in well illuminated shrubby clearings and grey dune habitats was completely different (Figure 4). As relationships of the pod width with the number of ovules, number of matured seeds, and pod width were weak (Figure 3), we suppose that the width of pods is affected by other factors that were not included in this study. The genetic diversity of *C. scoparius* populations [28,31] also could be among those factors determining pod width and length variation, similar as it is with frost tolerance and other traits between different genotypes [32]. 

### 4.3. Comparison of Reproductive Traits between Years

Irregular differences of the mean number of ovules per pod in different study years points to effects of environmental conditions in the previous year. However, irregularities were registered at two sites (i.e., Juodkrantė and Nagliai), despite them having almost identical meteorological conditions. Thus, we suppose that the different mean number of ovules between years at the same site could be influenced by the total fruit yield in the previous year. Several studies have shown that many plants develop less ovules in ovaries and set less and smaller fruits with fewer seeds in the year following one with high yield [85,86,87,88]. However, we cannot confirm this assumption because we did not evaluate the total fecundity of plants in different years of the study.

We also assume that the age of *C. scoparius* shrubs could be an important individual characteristic, affecting the number of ovules per pod. We assume that young generative individuals develop smaller numbers of ovules per pod than individuals at the peak of their generative phase. The relationship of increased seed number per fruit and general fecundity with the age of individuals has been confirmed by other studies [88,89,90] and possibly could be applied for the case of *C. scoparius*. However, this assumption would need to be verified by a separate study.

Comparing the mean number of matured seeds per pod between years showed that in individual sites, as well as in all sites pooled, it was significantly higher in 2017 than in 2016, except for the Juodkrantė site. The realised fecundity in three studied sites was also higher in 2017 than in 2016. The mean number of ovules per pod was different between the two years, however this does not explain the apparently different realised fecundity in the study period (35.9% in 2016 and 57.2% in 2017). Thus, this fact suggests an effect of other factors on the difference in the realised fecundity in different years. During the flowering season of *C. scoparius* in May of 2017, the amount of precipitation was lower than in 2016 [73,74]. Thus, we suppose that more abundant rainfall during the flowering season might affect the seed set rate by impeding flower pollination by insects. The results of some studies revealed that the activity of bumblebees, which are the most important pollinators of *C. scoparius*, is negatively associated with rainfall, humidity, and wind speed [91]. On sunny days with no rain, bumblebees visit flowers several times more frequently than on cloudy and rainy days [92]. Thus, the significantly higher amounts of precipitation in May of 2016 (37.7 mm) than in May of 2017 (6.3 mm) might have determined the effectiveness of flower pollination. As the flowering season of 2017 was dry, flower pollination was more intense and effective, and thus the number of developed seeds and the realised fecundity was significantly higher. Different percentages of ovule abortion and seed-set rate between years has been reported for *Bauhinia ungulata* L. suggesting different pollination success [93]. 

Differences in pod length and width between years at the same sites can possibly be explained by the different meteorological conditions during the pod development season. When pods of *Cytisus scoparius* develop most intensively in June and July, the total amount of precipitation was almost twice higher in 2017 than in 2016 (210.5 mm and 145.0 mm, respectively; [73,74]. The significant increase of pod length and width at the Nagliai site in 2017 could be related to the increased number of matured seeds as compared with 2016. The results of other studies show that a greater number of fertilised ovules facilitates production of phytohormones which, in turn, enhances fruit development [83,84]. However, pod length and width at the Valkininkai site was similar within the two years of the study, but the number of ovules, the number of matured seeds was significantly different between 2016 and 2017. This fact suggests that fruit size variation depends on an intricate combination of environmental (habitat type, light availability and intensity, precipitation, etc.) and internal factors affecting individual *C. scoparius* plants. 

## 5. Conclusions

This study was not able to unambiguously confirm the hypothesis that *C. scoparius* in the invasive range in Lithuania, in the southern part of the Boreal biogeographical region, may outperform plants in the native range as based on the analysed reproductive traits. The geographic location and the status of populations do not explain the intricate pattern of reproductive trait variation. Potential fecundity of *C. scoparius* in the native range was significantly higher than in the invaded range; however, realised fecundity was not significantly different between the native and invasive ranges. Pod length was similar in both ranges, whereas pod width was significantly greater in the invasive range than in the native range. Nevertheless, the obtained results suggest that the number of ovules per pod, number of matured seeds, and the size of pods are correlated with the type of habitats and local environmental conditions in the habitat. The meteorological conditions during the flowering season affect the success of flower pollination and significantly contribute to the outcome of realised fecundity. 

Even though the realised fecundity of *C. scoparius* in shaded woodland habitats is significantly lower than in open and semi-open habitats, seed production in woodlands is sufficient for supporting population stability and further spread and invasion, particularly in Scots pine stands of the southern part of the Boreal biogeographical region of Europe.

Although many studies on other species confirmed higher fecundity and overall better performance of plants in the invasive range than in the native range in other continents, this rule probably cannot be applied for intracontinental invasive species occurring in relatively close geographical regions to their native ranges. More studies on other intracontinental invasive species should be performed aiming to reveal whether the case of the reproductive performance of *C. scoparius* is an exception or a more general phenomenon. 

## Figures and Tables

**Figure 1 plants-11-02158-f001:**
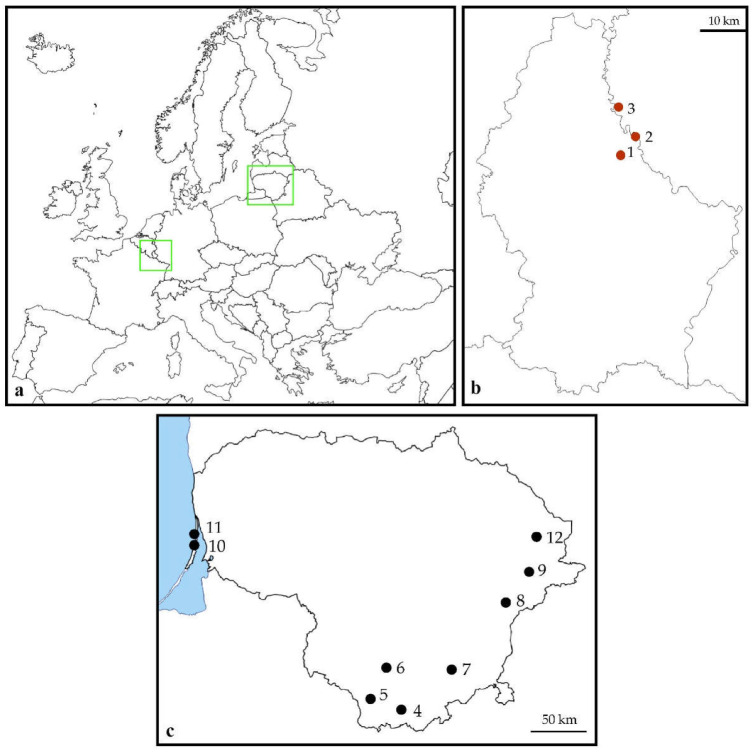
*Cytisus scoparius* study areas (green rectangles, (**a**) and locations of study sites in Luxembourg and Germany (**b**) and Lithuania (**c**). Red dots indicate survey sites in the native range and black dots in the invasive range. The numbering of the survey sites corresponds to the numbering in Table 1.

**Figure 2 plants-11-02158-f002:**
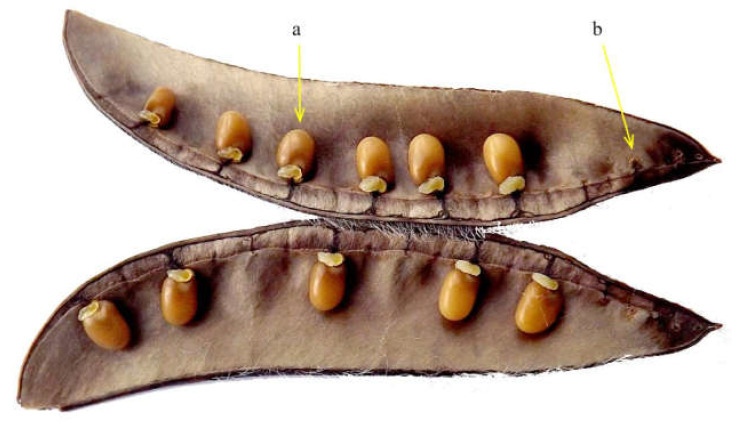
Opened pod of *Cytisus scoparius* with matured seeds (**a**) and aborted ovules (**b**).

**Figure 3 plants-11-02158-f003:**
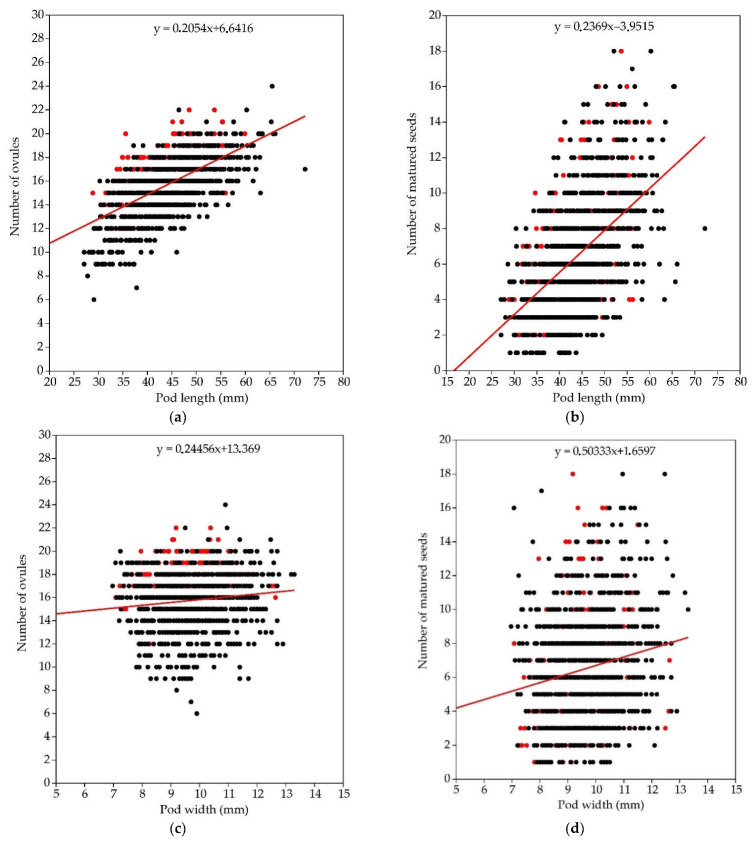
Relationship between pod length and number of ovules (**a**), pod length and number of mature seeds (**b**), pod width and number of ovules (**c**), and pod width and number of mature seeds (**d**). Red dots indicate data from the native range and black dots from the invasive range.

**Figure 4 plants-11-02158-f004:**
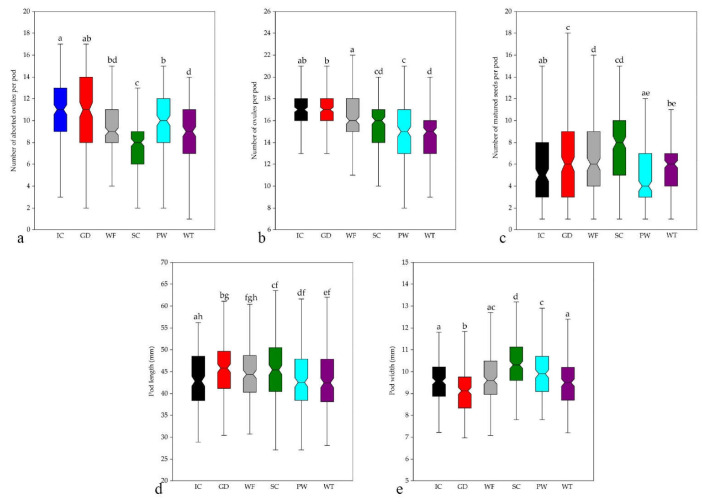
Comparison of the reproductive traits of *Cytisus scoparius* in the sites grouped by habitat type (n = 200 in each). different habitat types by the number of ovules per pod (**a**), number of matured seeds (**b**), number of aborted ovules (**c**), pod length (**d**), and pod width (**e**). Abbreviations of the habitat types: IC—inland cliffs, DG—grey dunes, WF—woodland fringes, SC—shrubby clearings, PW—Scots pine woodlands, WE—western taiga. Different letters indicate statistically significant differences of the trait mean values between habitat types according to the Tukey’s post-hoc pairwise comparisons.

**Table 1 plants-11-02158-t001:** Localities and habitats of the sampled *Cytisus scoparius* populations in the native range in Germany and Luxembourg and in the invasive range in Lithuania.

Number	Site Name	Country and Administrative Unit	Longitude (°N)	Latitude (°E)	Habitat
	* **Native range** *				
1	Walsdorf	Luxembourg, Bamberg distr.	49.92678	6.17394	Woodland fringe
2	Vianden	Luxembourg, Diekirch distr.	49.94018	6.20448	Inland cliff
3	Übereisenbach	Germany, Bitburg-Prüm distr.	49.99762	6.14206	Inland cliff
	* **Invasive range** *				
4	Grūtas	Lithuania, Druskininkai distr.	54.02997	24.06246	Shrubby clearing
5	Veisiejai	Lithuania, Lazdijai distr.	54.13799	23.67147	Woodland fringe
6	Meteliai	Lithuania, Lazdijai distr.	54.26898	23.75269	Shrubby clearing
7	Valkininkai	Lithuania, Varėna distr.	54.35036	24.71771	Pine woodland
8	Kabakėlis	Lithuania, Švenčionys distr.	55.01001	25.72869	Taiga woodland
9	Kaltanėnai	Lithuania, Švenčionys distr.	55.27716	26.12015	Taiga woodland
10	Nagliai	Lithuania, Neringa city	55.46504	21.08314	Grey dunes
11	Juodkrantė	Lithuania, Neringa city	55.52240	21.09606	Grey dunes
12	Didžiasalis	Lithuania, Ignalina distr.	55.56388	26.23861	Pine woodland

**Table 2 plants-11-02158-t002:** Mean values and standard deviations of reproductive traits of *Cytisus scoparius* at the studied sites in the native and invasive ranges.

Sites	Number of Ovules	Number of Matured Seeds	Number of Aborted Ovules	Pod Length (mm)	Pod Width (mm)
*Native range*					
Walsdorf	17.09 ± 1.54	5.88 ± 2.82	11.21 ± 2.77	43.81 ± 5.76	9.55 ± 0.85
Vianden	16.59 ± 1.89	5.58 ± 3.61	11.01 ± 3.36	42.53 ± 6.88	9.60 ± 1.16
Übereisenbach	17.41 ± 1.92	7.83 ± 3.13	9.58 ± 2.61	44.98 ± 5.78	9.46 ± 0.95
*Pooled native range*	17.03 ± 1.81	6.43 ± 3.34	10.60 ± 3.01	43.77 ± 6.22	9.54 ± 0.99
* **Invasive range** *					
Grūtas	16.89 ± 1.47	9.05 ± 2.70	7.77 ± 2.42	49.15 ± 7.49	10.44 ± 1.10
Veisiejai	15.05 ± 2.31	5.95 ± 3.36	9.14 ± 3.02	44.63 ± 7.23	10.01 ± 1.07
Meteliai	13.65 ± 2.25	6.34 ± 2.79	7.31 ± 2.29	42.62 ± 5.97	10.27 ± 1.04
Valkininkai	15.09 ± 2.47	4.72 ± 3.20	10.37 ± 2.96	44.00 ± 8.81	9.76 ± 1.14
Kabakėlis	14.71 ± 2.07	5.77 ± 2.00	8.94 ± 2.49	44.79 ± 6.75	9.58 ± 1.32
Kaltanėnai	14.07 ± 2.48	5.79 ± 3.25	8.28 ± 3.01	41.32 ± 7.43	9.49 ± 0.91
Nagliai	16.67 ± 1.88	6.78 ± 3,37	9.89 ±2.62	43.73 ± 6.74	9.27 ± 0.80
Juodkrantė	17.09 ± 1.87	9.44 ± 3.23	7.65 ± 2.80	47.08 ± 6.46	8.85 ± 1.35
Didžiasalis	14.58 ± 2.33	5.36 ± 2.60	9.22 ± 2.56	43.05 ± 6.52	10.21 ± 1.11
*Pooled invasive range*	15.31 ± 2.43	6.58 ± 3.24	8.73 ± 2.86	44.48 ± 7.41	9.76 ± 1.20

**Table 3 plants-11-02158-t003:** Comparison of reproductive traits of *Cytisus scoparius* at three sites in 2016 and 2017. Data includes mean values and standard deviation. Differences between different years at the same site were assessed by applying a t-test (*—*p* < 0.05, **—*p* < 0.01, ***—*p* < 0.001, n.s.—non-significant).

Sites	Year	Number of Ovules	Number of Matured Seeds	Number of Aborted Ovules	Pod Length (mm)	Pod Width (mm)
Valkininkai	2016	15.09 ± 2.47 ***	4.72 ± 3.20 ***	10.37 ± 2.96 ***	44.00 ± 8.8 ^n.s.^	9.76 ± 1.14 ^n.s.^
	2017	16.87 ± 1.66 ***	9.16 ± 3.24 ***	7.71 ± 2.35 ***	43.72 ± 5.68 ^n.s.^	9.50 ± 1.19 ^n.s.^
Juodkrantė	2016	17.09 ± 1.87 ^n.s.^	9.44 ± 3.23 ^n.s.^	7.65 ± 2.80 ^n.s.^	47.08 ± 6.46 ^n.s.^	8.85 ± 1.35 *
	2017	17.16 ± 1.76 ^n.s.^	10.15 ± 3.14 ^n.s.^	7.01 ± 2.86 ^n.s.^	47.81 ± 5.48 ^n.s.^	9.26 ± 0.84 *
Nagliai	2016	16.67 ± 1.88 *	6.71 ± 2.04 ***	9.96 ± 2.43 ***	43.73 ± 6.74 ***	9.27 ± 0.80 ***
	2017	16.06 ± 2.03 *	9.26 ± 2.77 ***	6.80 ± 2.21 ***	47.86 ± 5.92 ***	10.09 ± 0.94 ***
All sites polled	2016	16.28 ± 2.26 *	6.96 ± 3.46 ***	9.32 ± 2.98 ***	44.94 ± 7.54 **	9.29 ± 1.18 ***
	2017	16.70 ± 1.88 *	9.56 ± 3.10 ***	7.17 ± 2.51 ***	46.46 ± 6.00 **	9.62 ± 1.06 ***

**Table 4 plants-11-02158-t004:** Summary of the two-way ANOVA analyses of the relationships of *Cytisus scoparius* reproductive traits with the site (fixed factor) and year (random factor).

Traits and Factors	Sum of Squares	df	Mean Square	F	*p*
Number of ovules					
Site	135.79	2	67.89	0.89	0.409
Year	25.63	1	25.63	0.34	0.561
Interaction	151.64	2	75.82	19.64	<0.001
Within	2292.88	594	3.86		
Total	2605.94	599			
Number of matured seeds					
Site	834.61	2	417.30	2.40	0.092
Year	988.17	1	988.17	5.68	0.017
Interaction	347.84	2	173.42	19.76	<0.001
Within	834.61	594	8.90		
Total	7399.44	599			
Number of aborted ovules					
Site	297.48	2	148.74.	1.67	0.189
Year	695.53	1	695.53.	7.81	0.005
Interaction	178.01.	2	89.01	12.97	<0.001
Within	4075.48	594	6.86		
Total	5246.50	599			
Pod length					
Site	1292.29	2	646.14	2.82	0.060
Year	306.36	1	306.36	1.34	0.248
Interaction	458.50	2	229.25	5.46	0.004
Within	24,936.70	594	41.98		
Total	26,993.90	599			
Pod width					
Site	42.73	2	21.37	1.44	0.237
Year	13.94	1	13.94	0.94	0.332
Interaction	29.59	2	14.79	13.90	<0.001
Within	632.36	594	1.06		
Total	718.62	599

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
