# Peer review of "Do Reproductive Traits of Invasive Populations of Scotch Broom, *Cytisus scoparius* (Fabaceae), Outperform Native Populations?"

_plants, 2022, doi:10.3390/plants11162158_

Round 1

Reviewer 1 Report

The authors of this study investigated whether the reproductive traits of Scotch broom (Cytisus scoparius) perform better in the alien invasive range in Europe compared to its native range.

The title needs some rephrasing to sound good and read well; therefore, a change is suggested.

The abstract is good and only a few details should be corrected.

The introduction is well-focused on the topic and it presents an overview of current knowledge. Some minor shortcomings are indicated and suggestions are made with sticker notes in the pdf file to improve the quality of this part. The aim of the study needs some rephrasing, especially the questions made (see suggestions).

According to the journals' guidelines, the section Material and Methods appears after the Results and the Discussion sections and prior to the Conclusions. The authors should correct it during the revision process and they should re-enumerate all references after the introduction part.

In Material and Methods (but also in Results and Discussion), some inappropriate terms need to be substituted with concise ones (e.g., well-illuminated); these are indicated with sticker notes in the pdf file. In this section, a map outlining the native range and alien range of the studied species would fit well.

It is not clear whether the data taken from the Lithuanian Meteorological Yearbook cover also the studied sites in Germany and Luxembourg? Please, define.

The results and the discussion sections are the best parts of this manuscript in terms of elaboration and language editing. The authors present and discuss the findings of this study clearly and concisely, using a rich set of references adequately. A few minor shortcomings are indicated for improvement and suggestions are made with sticker notes in the pdf file.

The conclusions of this study are sound and fair, outlining the findings and their relative significance.

Latin names and statistical p should be carefully italicized throughout the manuscript.

The authors should check again ALL the journal abbreviations and they should ensure that the official journal abbreviations are provided with no improvisations (see sticker notes in the attached pdf file). Some corrections are also suggested in the list of references.

Although this manuscript is well-written it suffers from several scattered linguistic imperfections indicated with dozens of sticker notes in the pdf file attached. However, all these shortcomings taken together can be easily elaborated on during the revision process.

Author Response

Responses to the reviewer's comments on the manuscript

Do Invasive Populations of Scotch Broom, Cytisus scoparius (Fabaceae), Outperform Native Populations by their Reproductive Traits?

 We are very grateful to the reviewers for their analysis and evaluation of the manuscript. We have considered the comments and suggestions expressed and have made corrections to the manuscript accordingly.

 The editorial comments, with which we fully agree, are not discussed further. Corrections have been made to the text using the track change function. Other comments that are open to discussion or have been considered with some reservations are discussed and clarified following paragraphs in the review.

REVIEWER 1

  1. According to the journals' guidelines, the section Material and Methods appears after the Results and the Discussion sections and prior to the Conclusions. The authors should correct it during the revision process, and they should re-enumerate all references after the introduction part.

It is recommended that the Materials and Methods section be placed after the Discussion section in the Journal, but we believe that this section should logically precede the Results section so that the reader has immediate access to the Materials and Methods. A review of the articles already published in Plants shows that in some of them this section is placed between the Introduction and the Results (e.g., https://doi.org/10.3390/plants11151990). For the time being, therefore, we are leaving the Materials and Methods section as it was, and if the Editorial team strongly recommends the move, this technical change can be easily made.

  1. It was suggested to modify the title of the paper.

The title of the paper corrected following suggestion: Do Reproductive Traits of Invasive Populations of Scotch Broom, Cytisus scoparius (Fabaceae), Outperform Native Populations?

  1. In Material and Methods (but also in Results and Discussion), some inappropriate terms need to be substituted with concise ones (e.g., well-illuminated); these are indicated with sticker notes in the pdf file. In this section, a map outlining the native range and alien range of the studied species would fit well.

Terms in the text have been consistently revised and changed according to the reviewer's suggestions.

A precise range map for the species is not available and a separate large-scale survey would be required to produce reliable map of the native range. In particular, the northern and eastern boundaries of the species' natural range are uncertain. In Poland, the limit of the species' native range is thought to be south of Warsaw. Therefore, the range description is left to the text. We expect that the revised native and anthropogenic ranges of the species will be published in the next volume of Atlas Florae Europaeae, maybe in 2023.

  1. It is not clear whether the data taken from the Lithuanian Meteorological Yearbook cover also the studied sites in Germany and Luxembourg? Please, define.

Meteorological data were used only for the comparison of the three populations surveyed in 2016 and again in 2017. All of them were from Lithuania. We did not use meteorological data from other areas because they are not sufficiently accurate, i.e., the locations surveyed are too far from the meteorological stations. Populations close to the weather stations (less than 5–15 km from the survey site) were specifically selected for the two-year study. We believe that the methodology clearly states that temperature and precipitation data were used to compare the three populations.

  1. The results and the discussion sections are the best parts of this manuscript in terms of elaboration and language editing. The authors present and discuss the findings of this study clearly and concisely, using a rich set of references adequately. A few minor shortcomings are indicated for improvement and suggestions are made with sticker notes in the pdf file.

Editorial changes have been made to the text in line with the reviewer's comments.

  1. Latin names and statistical p should be carefully italicized throughout the manuscript.

Latin names and p-values have been checked and, where missing, their font has been replaced by italics.

  1. The authors should check again ALL the journal abbreviations and they should ensure that the official journal abbreviations are provided with no improvisations (see sticker notes in the attached pdf file). Some corrections are also suggested in the list of references.

Abbreviations of journal titles checked. We did not find standard abbreviations for some journal titles, so such titles have been left unabbreviated to avoid confusion.

Reviewer 2 Report

This manuscript compared populations of Scotch broom in native and invasive areas within Europe to determine if there were differences in seed production and seed pod morphology. The authors found differences among populations found in different habitats but these differences were not directly linked to native versus invasion areas. Although the study system is interesting and the questions are of interest in controlling invasive species, I have several concerns about the manuscript in its current form. I have placed specific comments below that I hope will benefit the authors.

Abstract:

1. The authors should define the location of the native area in the abstract. How far away from the populations in Lithuania does Scotch broom occur? Also, define what is meant by potential and realized fecundity.

2. Ln 24-37: This is a very broad statement given that only a single species is examined in the study. Whether or not other invasive plants follow this pattern is not presented. Also after having read the manuscript, the results don’t support such a strong conclusion.

Introduction:

1. A definition of what is meant by alien and invasive is needed. How exactly do the authors define an invasive species?

2. Ln 56-57 and 61-63 seem to contradict one another. 29% seems like a large number of species. Could the authors rewrite for clarity?

3. Ln 57-60: what did other studies find when they compared within the same continent? Did they find differences in ecological performance, genetic diversity, or population structure?

4. Ln 66 – 70: the few sentences about how the species came to be in Lithuania suggest that it would not be invasive if humans weren’t continuing to introduce it into Lithuania. How did the authors/or other researchers determine that Scotch broom is an invasive (e.g., outcompeting native species etc…) in that area?

5. A map is needed to help clarify many components of the manuscript. Where are the locations in Lithuania and other countries where plants were sampled? Where does the Scotch broom occur natively? It is difficult to interpret the results without a map.

6. Ln 90: what does “merely investigated” mean? How does this differ from being studied?

7. Ln 105-106: what are Baker’s postulates? This seems important to the paper but is not explained.

Materials and Methods:

1. Do the authors have any information about where the plants came from that were introduced by humans into Lithuania? Are they all from the same area? Did they come from Germany or Luxembourg? How might genetic relatedness among populations effect the results of the study? How much does genetic relatedness effect seed production and pod morphology?

2. Ln 152: no analyses were run that included the meteorological information; not in the methods or results.

3. Ln 163: how did the authors make sure that they were measuring the same location for the width of each pod? Was it the widest part? In addition, there isn’t an explanation as to why pod morphology is being examined. What does it provide that cannot be learned from studying seed production?  

4. Ln 175: Tukey’s HSD post hoc test?

5. Ln 176: how were data sets unequal? Sample size? Variance? Also, parametric analyses require homogeneity of variance in most instances, was this tested for?

6. Ln 179-180: this sentence is somewhat confusing. Were comparisons made between years for each variable that was measured?

7. Correlations between variables were assessed but the analysis was not included in the methods.

8. Why were the response variables compared to one another? It is not surprising that these variables would be highly correlated given that they are not independent of one another. It is not clear how this helped in answering the questions posited in the introduction.

9. After having read the entire manuscript, I think that a much better way to analyze the data would be to perform a multiple regression that included both categorical and continuous predictor variables. This would allow the authors to take into account the effects of year, weather, native vs invasive range, location sampled, and habitat type. The authors could also use an information theoretic approach to find the model that best fits the variation in their response variables. I think this would be much more informative than the analyses that were included in the manuscript.

10. Ln 185: what is obtained from reference data?

Results:

1. Ln 191: how was the significant variation determined in ovules per pod?

2. Tables are fine but figures would be much more informative to show the relationships among locations. Possibly boxplots? This would be a better representation of the data for all ANOVA comparisons.

3. Ln 199: what does population origin mean?

4. Ln 220: How did the authors determine that differences among populations were due to differences in habitat types?

5. Ln 223-224: The authors should be careful when saying that a mean value is lower when one mean does not differ significantly from another mean.

6. Ln 241-242: “realization of potential fecundity” is confusing. I would suggest that they remove the realization part given the possible confusion with realized and potential fecundity.

7. Ln 261: how many is most?

8. This was mentioned previously but it is unclear why pod morphology was measured in relation to the manuscript’s goals.

9. Fig. 2: why did the authors combine native and invasive samples in the plots?

10. Ln 288: habitat type comparisons were not included in the methods

11. Table 3: how many locations from invasive and native are in each habitat type? Which locations are associated with which habitats? How many individuals are from each habitat? Is it valid to combine the native and invasive samples? How much of an effect does sample location have on these results? This could be teased out using a multiple regression.

12. Ln 372: should be mature not matures

Discussion:

1. How well does seed production evaluate fecundity in this species? Given that there is a large seedbank…

2. Ln 408: what does Baker postulate?

3. Ln 412: meteorological variables were not included in the methods or results

4. Ln 423: range not rage

5. Ln 427-439: these comparisons should be stated in the introduction, methods, and results

6. Ln 438: flower pollination was not included in the study.

7. Ln 440-450: It is unclear why pod length or width is important to the questions put forward in the study.

8. Ln 462: light availability was not included in the study.

9. Ln 487-490: why couldn’t genetic differences among locations explain differences among seed production and pod morphology?

10. Ln 523-524: Did the authors attempt to minimize the effects of different ages among plants by sampling plants of similar size?

11. Ln 535: why not include comparisons for precipitation levels in the methods or results?

12. Ln 552-556: amounts of precipitation could be included in a multiple regression analysis as suggested above.

Author Response

Responses to the reviewer's comments on the manuscript

 Do Invasive Populations of Scotch Broom, Cytisus scoparius (Fabaceae), Outperform Native Populations by their Reproductive Traits?

 We are very grateful to the reviewers for their analysis and evaluation of the manuscript. We have considered the comments and suggestions expressed and have made corrections to the manuscript accordingly. The comments that are open to discussion or have been considered with some reservations are discussed and clarified following paragraphs in the review.

REVIEWER 2

  1. The authors should define the location of the native area in the abstract. How far away from the populations in Lithuania does Scotch broom occur? Also, define what is meant by potential and realized fecundity.

Sentence added in the Introduction “The invasive populations of Cytisus scoparius in Lithuania are located approximately 300 km north of their native range boundary in Poland.” We have not included this sentence in the abstract because it would greatly extend the abstract and exceed the word count recommended by the editors.

The meaning of potential and realised fecundity is defined in the first paragraph of introduction: “The potential fecundity in plants usually is estimated as the number of developed ovules, whereas realised fecundity is defined as the number of matured seeds or ratio between the number of ovules and matured seeds [4-6].” We consider the inclusion of this definition in the abstract to be superfluous.

  1. Ln 24-37: This is a very broad statement given that only a single species is examined in the study. Whether or not other invasive plants follow this pattern is not presented. Also after having read the manuscript, the results don’t support such a strong conclusion.

The Introduction describes transcontinental invasions, whereas there are no studies on intracontinental invasions to compare with. For this reason, this study was undertaken to determine whether the same rules apply to intracontinental invasions. The end of the Abstract does not contradict the results obtained, but confirms them, i.e., it states that the superiority of invasive populations over native populations is not confirmed. (“Although many studies on other species have confirmed higher fecundity and overall better performance of plants in the invasive area than in the native area on other continents, this rule probably cannot be applied for intracontinental invasive species occurring in relatively close geographical regions to their native areas.”)

Introduction:

  1. A definition of what is meant by alien and invasive is needed. How exactly do the authors define an invasive species?

Definition of alien and invasive species is presented trough citation of references with exhaustive definitions:

  1. Richardson D.M.; Pyšek P.; Rejmanek M.; Barbour M.G.; Panetta D.M.; West C.J. Naturalization and invasion of alien plants: Concepts and definitions. Divers. Distrib. 2000, 6, 93–107. https://doi.org/10.1046/j.1472-4642.2000.00083.x
  2. Pyšek P.; Richardson D.M. Traits associated with invasiveness in alien plants: where do we stand? In: Ecological Studies ed. Nentwig W. 2007, 97–125. https://doi.org/10.1007/978-3-540-36920-2_7

It would, in our opinion, be an unnecessary extension of the text to repeat definitions that are cited and well known in the literature.

  1. Ln 56-57 and 61-63 seem to contradict one another. 29% seems like a large number of species. Could the authors rewrite for clarity?

The indicated texts are not contradictory, as these paragraphs deal with completely different subjects. It is claimed that there are many alien species in Europe that originate from the same continent, but only a small proportion of them are listed as invasive.

  1. Ln 57-60: what did other studies find when they compared within the same continent? Did they find differences in ecological performance, genetic diversity, or population structure?

Studies that have been carried out on intracontinental invasive species have shown that a large proportion of the populations are the result of multiple introductions, but population performance has not been analysed. Therefore, the results of these studies are not discussed further as they are not directly relevant to this study. With this statement we wanted to show that there are very few studies on intercontinental invasions.

  1. Ln 66–70: the few sentences about how the species came to be in Lithuania suggest that it would not be invasive if humans weren’t continuing to introduce it into Lithuania. How did the authors/or other researchers determine that Scotch broom is an invasive (e.g., outcompeting native species etc…) in that area?

If humans had not introduced Cytisus scoparius, it is highly likely that it would not exist in Lithuania, or would not have become invasive yet. Negative impacts of the species on local habitats and biodiversity have been evaluated in areas where the invasion of the species is recent. Several populations of Pulsatilla patens are reported to have been lost due to the invasion of this species, and populations of Thesium ebracteatum have been severely affected. The most significant mechanism of adverse effects is the enrichment of the soil with nitrogen (eutrophication of the habitat).

  1. A map is needed to help clarify many components of the manuscript. Where are the locations in Lithuania and other countries where plants were sampled? Where does the Scotch broom occur natively? It is difficult to interpret the results without a map.

The geographical coordinates of the locations are given in Table 1, which make it easy to identify the research sites. At the request of the reviewer, we have attached an additional map with the survey locations (Figure 1). A precise range map for the species is not available and a separate large-scale survey would be required to produce reliable map of the native range. In particular, the northern and eastern boundaries of the species' natural range are uncertain. In Poland, the limit of the species' native range is thought to be south of Warsaw. Therefore, the range description is left to the text. We expect that the revised native and anthropogenic ranges of the species will be published in the next volume of Atlas Florae Europaeae, maybe in 2023.

  1. Ln 90: what does “merely investigated” mean? How does this differ from being studied?

This means “almost neglected”, as the publications only mention this phenomenon but nowhere go into detail.

  1. Ln 105-106: what are Baker’s postulates? This seems important to the paper but is not explained.

Baker [69. Baker H.G. Characteristics and modes of origin of weeds. In The genetics of colonising species. Baker, H.G and Stebbins, G.L eds; New York, Academic Press: 1965 pp. 147–168.] postulated that alien species in the invasive range perform better than in the native range. In our opinion, the essence of the postulate is clear from the hypothesis, and the postulate itself is widely discussed and well known to those studying invasion ecology.

Materials and Methods:

  1. Do the authors have any information about where the plants came from that were introduced by humans into Lithuania? Are they all from the same area? Did they come from Germany or Luxembourg?

The plants are known to have been introduced from Germany at the end of the 19th century, but the fate of these populations is unknown, as Cytisus scoparius is not currently found in these areas. The origin of the plants introduced in the middle of the 20th century is unknown, as the introduction was not documented during the Soviet period. The introduction of the species to Lithuania has been discussed in the literature cited [39. Taura L.; Gudžinskas Z. Life stages and demography of invasive shrub Cytisus scoparius (Fabaceae) in Lithuania. Botanica 2020, 26, 1–14. https://doi.org/10.2478/botlit-2020-0001], so it is not appropriate to repeat it in this paper.

1a. How might genetic relatedness among populations effect the results of the study? How much does genetic relatedness effect seed production and pod morphology?

The results of the study and the information in the literature do not suggest that genetic relationships can have a significant effect on the reproductive traits studied, although this cannot be ruled out either. All the results show that habitat conditions have a significant effect on reproductive performance. Seed production is most affected by meteorological conditions during the flowering period.

  1. Ln 152: no analyses were run that included the meteorological information; not in the methods or results.

Information on the temperature and precipitation at the study sites and in study years was added to the Materials and Methods chapter.

  1. Ln 163: how did the authors make sure that they were measuring the same location for the width of each pod? Was it the widest part? In addition, there isn’t an explanation as to why pod morphology is being examined. What does it provide that cannot be learned from studying seed production?

It is clearly stated in the methodology section that the width of the pod is measured at its centre. The width of the pods was in most cases a constant value, varying by tenths of a millimetre. The centre of the pod is the widest point. Pod morphology has been assessed as one of the reproductive traits. Data on pod morphology can be used for further studies on reproductive performance of the species. It has also been found that the length of the pods depends more on the number of matured seeds than on the number of ovules.

  1. Ln 175: Tukey’s HSD post hoc test?

Corrected.

  1. Ln 176: how were data sets unequal? Sample size? Variance? Also, parametric analyses require homogeneity of variance in most instances, was this tested for?

The datasets were uneven because 3 populations (n = 300) from the native range and 9 populations (n = 900) from the invasive range were sampled. In such cases, non-parametric methods are applied to compare the datasets. The text was corrected.

Homoskedacity of the data was estimated applying Levene’s test and the results of the test confirmed homogeneity of variance (the smallest p value was 0.085 for the number of seeds per pod). The sentence about evaluation of homoskedacity was added to the text.

  1. Ln 179-180: this sentence is somewhat confusing. Were comparisons made between years for each variable that was measured?

The sentence was corrected.

  1. Correlations between variables were assessed but the analysis was not included in the methods.

Correlation between variables are the results of bivariate linear regression analysis, which is described in the methods. “Relationships between traits (the number of ovules, number of matured seeds, pod length and width) were evaluated applying generalized linear model and bivariate linear regression analysis.”

  1. Why were the response variables compared to one another? It is not surprising that these variables would be highly correlated given that they are not independent of one another. It is not clear how this helped in answering the questions posited in the introduction.

The correlation analysis between the response variables was carried out purposely to determine the relationship between the number of ovules and the number of matured seeds. The analysis showed that the correlation is relatively weak, suggesting that the number of matured seeds per pod is determined by factors other than the number of ovules.

  1. After having read the entire manuscript, I think that a much better way to analyse the data would be to perform a multiple regression that included both categorical and continuous predictor variables. This would allow the authors to take into account the effects of year, weather, native vs invasive range, location sampled, and habitat type. The authors could also use an information theoretic approach to find the model that best fits the variation in their response variables. I think this would be much more informative than the analyses that were included in the manuscript.

Multiple regression methods were used to analyse the data but did not work. The impact of meteorological conditions in all areas was not analysed, as the information available is very imprecise. Only three populations, which are close to meteorological stations and where the exact rainfall amounts are known, have been targeted for meteorological impact assessment. When the population is more than 30 km away from the weather station, the recorded rainfall amounts do not reflect reality at all. Taking all these factors into account, other methods of analysis have been adopted which show the differences much better and are much more accurate.

  1. Ln 185: what is obtained from reference data?

Data on the length of pods and number of ovules per pod from Spain published in the literature, were used for comparison. We have revised the text.

Results:

  1. Ln 191: how was the significant variation determined in ovules per pod?

The text has been corrected and the term significantly has been replaced by the neutral term considerably.

  1. Tables are fine, but figures would be much more informative to show the relationships among locations. Possibly boxplots? This would be a better representation of the data for all ANOVA comparisons.

We have used box-plot graphs instead of the Table to compare results between habitats. However, we believe that tables are very important as they provide accurate results, whereas graphs often do not allow accurate reproduction of numerical values. Accurate numerical values are important when we want to compare the results of similar studies with each other. We have therefore moved the comparison table of the analysed traits to the Appendix.

  1. Ln 199: what does population origin mean?

Population origin refers to whether it is native or invasive.

  1. Ln 220: How did the authors determine that differences among populations were due to differences in habitat types?

This is not stated, but it is assumed that there may be such an influence (Ln 220: “but is suggested to be dependent on habitat type”). The influence of habitats is dealt with in the next sub-section.

  1. Ln 223-224: The authors should be careful when saying that a mean value is lower when one mean does not differ significantly from another mean.

Therefore the text emphasises that the mean values differ, but there is no significant difference between them.

  1. Ln 241-242: “realization of potential fecundity” is confusing. I would suggest that they remove the realization part given the possible confusion with realized and potential fecundity.

We revised the sentence to avoid ambiguity.

  1. Ln 261: how many is most?

“Most” represents more than half of the total entities. Reference is made to the table in the Appendix for the exact estimated differences.

  1. This was mentioned previously but it is unclear why pod morphology was measured in relation to the manuscript’s goals.

Pod morphology was analysed aiming to answer, “What is the relationship between potential and realised fecundity and the size of pods?”

  1. Fig. 2: why did the authors combine native and invasive samples in the plots?

We have put the native and invasive populations together in the graphs to show the similarities in the distribution of the parameters considered.

  1. Ln 288: habitat type comparisons were not included in the methods

Section of Statistical analyses was supplied with information about comparison of habitat types.

  1. Table 3: how many locations from invasive and native are in each habitat type? Which locations are associated with which habitats? How many individuals are from each habitat? Is it valid to combine the native and invasive samples? How much of an effect does sample location have on these results? This could be teased out using a multiple regression.

The habitat type of each population studied is indicated in Table 1. This gives a clear indication of the populations studied. As indicated in the methodology, 50 individuals were sampled per site (2 pods per individual). Thus, 100 individuals and 200 pods from each habitat type were surveyed. This information is clearly stated in the methodology. Since the habitats concerned are the same, it is logical to combine the data, especially since the aim is to answer the question of the effect of habitat type on reproductive traits.

  1. Ln 372: should be mature not matures

Corrected.

Discussion:

  1. How well does seed production evaluate fecundity in this species? Given that there is a large seedbank…

The large seed bank is not only a result of the longevity of the seeds, but also of the high seed production. This leads to the conclusion that even in the western taiga forests, a large seed bank can be created, ensuring the long-term stability of the invasive population.

  1. Ln 408: what does Baker postulate?

Baker [69] postulated that alien species in the invasive range perform better than in the native range. We believe that it is not necessary to repeat well-known statements, but that it is sufficient to cite sources.

  1. Ln 412: meteorological variables were not included in the methods or results

Information on the temperature and precipitation at the study sites and in study years was added to the Materials and Methods chapter.

  1. Ln 423: range not rage

Corrected.

  1. Ln 427-439: these comparisons should be stated in the introduction, methods, and results

Information on the results of the studies from Spain is given in the introduction and the comparison is described in the methods. We do not mention them in the results as they are part of the discussion, not the results.

  1. Ln 438: flower pollination was not included in the study.

We have not studied pollination of flowers, but we believe that we can and must discuss the results of studies by other authors that explain the phenomena studied.

  1. Ln 440-450: It is unclear why pod length or width is important to the questions put forward in the study.

Pod morphology was analysed aiming to answer, “What is the relationship between potential and realised fecundity and the size of pods?”

  1. Ln 462: light availability was not included in the study.

Light availability was included in the study as habitat characteristic. Characteristics of habitats were added in the Annex (Table 1A).

  1. Ln 487-490: why couldn’t genetic differences among locations explain differences among seed production and pod morphology?

It would be hard to believe that genetic reasons for realised productivity should be sought when there are obvious ecological reasons. And if that can be the case, then a separate study is needed. A single study cannot answer the many questions that arise when addressing issues that have been little studied. Differences in pod width could be due to genetic causes, but the aim of our study was to compare potential and realised fecundity in the native and invasive range.

  1. Ln 523-524: Did the authors attempt to minimize the effects of different ages among plants by sampling plants of similar size?

Cytisus scoparius has a low correlation between age and size, as some of the branches die back periodically. Stem diameter and age are more correlated. This is described in the paper on age composition of populations [39. Taura L.; Gudžinskas Z. Life stages and demography of invasive shrub Cytisus scoparius (Fabaceae) in Lithuania. Botanica 2020, 26, 1–14. https://doi.org/10.2478/botlit-2020-0001]. For the sampling of pods, we selected healthy and well-developed individuals of approximately the same size.

  1. Ln 535: why not include comparisons for precipitation levels in the methods or results?

Information on the temperature and precipitation at the study sites and in study years was added to the Materials and Methods chapter.

  1. Ln 552-556: amounts of precipitation could be included in a multiple regression analysis as suggested above.

Amount of precipitation was included in the two-way ANOVA analysis. Multiple regression analysis was tested; however, this method of analysis was rejected because the results showed no patterns.

Reviewer 3 Report

Author has reported the manuscript for the possible publication in plant. The idea is good but sample size is less. Following are my main suggestion. 

1. What is native base area of , Cytisus scoparius?

2. The authors should display form of sampling Study Sites.

3.  The study is too old. Material for the study on the reproductive traits of Cytisus scoparius was collected 134 from July to August of 2016. what is reason to delay ?

4. The Authors should add more picture from different sampling area to make study more effective. 

5. What is main reason to work only on ovules? why not other reproductive part. 

Author Response

Responses to the reviewer's comments on the manuscript

 Do Invasive Populations of Scotch Broom, Cytisus scoparius (Fabaceae), Outperform Native Populations by their Reproductive Traits?

 We are very grateful to the reviewers for their analysis and evaluation of the manuscript. We have considered the comments and suggestions expressed and have made corrections to the manuscript accordingly. The comments that are open to discussion or have been considered with some reservations are discussed and clarified following paragraphs in the review.

REVIEWER 3

Comments and Suggestions for Authors

Author has reported the manuscript for the possible publication in plant.

The idea is good but sample size is less. Following are my main suggestion.

In our opinion, 12 populations is a sufficiently large sample size and represents the most important habitats for the species.

  1. What is native base area of Cytisus scoparius?

The native area of Cytisus scoparius is described in the chapter Study species (2nd paragraph).

  1. The authors should display form of sampling Study Sites.

The sampling methodology is described in detail. All measurements and calculations of seeds and ovules are also described. We do not consider it appropriate to include a technical sampling protocol in the article, which will not provide any additional information and will only increase the length of the article.

  1. The study is too old. Material for the study on the reproductive traits of Cytisus scoparius was collected from July to August of 2016. what is reason to delay ?

The surveys were carried out in 2016-2017 and were part of a larger study. As this project did not receive external funding, the preparation of the paper was delayed. Nevertheless, the data and results are not outdated, in the same way as the results of research carried out 20 or 30 years ago and published in articles are relevant.

  1. The Authors should add more picture from different sampling area to make study more effective.

Due to the large size of the article, we did not hesitate to expand it further by adding habitat photos. If the reviewer strongly recommends the addition of photographs, the article could be expanded, although in general the photographs do not show significant differences in habitat. The habitat characteristics are additionally attached in Appendix A.

  1. What is main reason to work only on ovules? why not other reproductive part. 

In addition to ovules, the paper also examines other reproductive traits of Cytisus scoparius, such as the number of mature seeds, pod length and width, and the reasons behind the realised fecundity. Since the plant reproduces only by seeds, other reproductive traits that are very important for determining invasiveness are simply not available.